# Systematic Approach to Developing Splice Modulating Antisense Oligonucleotides

**DOI:** 10.3390/ijms20205030

**Published:** 2019-10-11

**Authors:** May T. Aung-Htut, Craig S. McIntosh, Kristin A. Ham, Ianthe L. Pitout, Loren L. Flynn, Kane Greer, Sue Fletcher, Steve D. Wilton

**Affiliations:** 1Centre for Molecular Medicine and Innovative Therapeutics, Murdoch University, Perth, WA 6150, AustraliaC.McIntosh@murdoch.edu.au (C.S.M.); Kristin.Ham@murdoch.edu.au (K.A.H.); I.pitout@murdoch.edu.au (I.L.P.); Loren.Flynn@murdoch.edu.au (L.L.F.); k.greer@murdoch.edu.au (K.G.); s.fletcher@murdoch.edu.au (S.F.); 2Perron Institute for Neurological and Translational Science, Centre for Neuromuscular and Neurological Disorders, The University of Western Australia, Perth, WA 6009, Australia

**Keywords:** antisense oligonucleotide, splice modulation, 2′-*O*-Methyl, transfection

## Abstract

The process of pre-mRNA splicing is a common and fundamental step in the expression of most human genes. Alternative splicing, whereby different splice motifs and sites are recognised in a developmental and/or tissue-specific manner, contributes to genetic plasticity and diversity of gene expression. Redirecting pre-mRNA processing of various genes has now been validated as a viable clinical therapeutic strategy, providing treatments for Duchenne muscular dystrophy (inducing specific exon skipping) and spinal muscular atrophy (promoting exon retention). We have designed and evaluated over 5000 different antisense oligonucleotides to alter splicing of a variety of pre-mRNAs, from the longest known human pre-mRNA to shorter, exon-dense primary gene transcripts. Here, we present our guidelines for designing, evaluating and optimising splice switching antisense oligomers in vitro. These systematic approaches assess several critical factors such as the selection of target splicing motifs, choice of cells, various delivery reagents and crucial aspects of validating assays for the screening of antisense oligonucleotides composed of 2′-*O*-methyl modified bases on a phosphorothioate backbone.

## 1. Introduction

Splice switching antisense oligonucleotides (AOs) are gaining interest as therapeutics for a wide variety of inherited and acquired diseases, with the approvals of *Eteplirsen* and *Nusinersen* to treat Duchenne muscular dystrophy and spinal muscular atrophy, respectively [1,2,3]. Antisense oligonucleotides are short, synthetic nucleic acids or analogues, typically 18 to 30 nucleotides in length that can specifically anneal to a complementary DNA or RNA sequence via Watson–Crick base-pairing [4]. Depending upon the nature of the AO (modified bases and backbone chemistry), these compounds can be designed to alter gene expression through several distinct mechanisms; including, but not limited to, RNase H mediated mRNA degradation [5], induction of RNA silencing [6], translation blockade [7], suppression of miRNA action, and modulation of pre-mRNA splicing [8]. The specific mechanism induced is determined by the AO chemistry as well as the annealing location in the gene transcript. Here, we specifically focus on AO design and delivery to modulate pre-mRNA processing, by redirecting splice site/motif selection to promote either specific exon skipping or retention in a target pre-mRNA [9]. We also provide reference AOs optimised to modulate ubiquitously expressed human gene transcripts that may be employed as controls to monitor the efficiency of transfection, RNA extraction, and RT-PCR amplification.

Splice modulating AOs are designed to anneal to elements within or flanking an exon and influence its recognition by the spliceosome so that the exon is preferentially retained or excised from the mature mRNA as required. Redirection of splicing is presumably a consequence of preventing positive (enhancer) or negative (silencer) splicing factors from recognising enhancer or silencer elements in the pre-mRNA transcript. Steric hindrance at these sites alters the recognition of normal splice sites by the splicing machinery and leads to alternative selection of exons or intronic sequences in the targeted transcript [10]. Web-based tools such as SpliceAid 2 [11], Human Splicing Finder [12], and RegRNA [13] have facilitated the prediction of potential splice factor motifs in any given sequence. Bioinformatics can contribute to the design of splice switching AOs [14,15]. While it is relatively straightforward to target AOs to predicted enhancer or silencer motifs, or to confirmed splice donor or acceptor sites [14,15,16], this approach does not consistently yield effective splice altering AO sequences. Although the original AO that induced specific dystrophin exon 23 skipping in *mdx* mouse muscle was directed to the donor splice site [17], AOs targeting the same coordinates of the human dystrophin transcript were completely ineffective [18]. Similarly, targeting the human dystrophin exon 51 donor splice site with AOs of different lengths and chemistries did not induce any exon skipping. Ultimately, identifying target domains within a pre-mRNA that influence splicing and then refining AO design through micro-walking must be done empirically. 

To date, our laboratory has screened over 5,000 AOs directed at numerous gene transcripts, linked to genetic diseases that may be potentially amenable to a splice intervention therapy. In addition, we are also exploring non-productive splicing to downregulate expression of selected gene transcripts through inducing non-functional isoforms by either excising exons encoding crucial functional domains or disrupting the reading frame. Consequently, we have developed general guidelines that are efficient and effective in developing biologically active splice switching antisense oligomers.The pre-mRNA sequence is interrogated by one or more in silico prediction programs to identify potential splice enhancer or silencer motifs.Antisense oligonucleotides, typically 20 to 25 mers, are designed to anneal to the target motifs and synthesised as 2′-*O*-methyl (2-OMe) modified bases on a phosphorothioate (PS) backbone.The test compounds are complexed with cationic liposome preparations and transfected into cells.After incubation, total RNA is extracted and the target transcript is amplified using RT-PCR to assess differences in pre-mRNA processing, with and without AO treatment.Oligomers shown to induce the desired changes in pre-mRNA processing are further refined by micro-walking around the annealing site and/or altering AO length.Transfection studies over a range of concentrations are performed to identify compound(s) that modify splicing in a dose-dependent manner, and at the lowest concentration.

Oligomers composed of 2-OMe PS can be either sourced commercially from a variety of suppliers or, in our case, synthesised in-house on a nucleic acid synthesiser such as an Expedite 8909 or Akta OligoPilot plus 10. The 2-OMe PS AOs are excellent research tools as they are relatively easily transfected into many different cultured cells with the aid of cationic lipoplexes such as Lipofectamine™ 3000, Lipofectin™, and Lipofectamine™ 2000. However, when subsequent protein or functional studies are required, the optimised sequences are generally synthesised as phosphorodiamidate morpholino oligomers (PMOs), as this chemistry offers more efficient and sustained splicing modification and protein isoform expression without non-specific effects [19]. The PMO chemistry is well tolerated both in vitro and in vivo, even at relatively high dosages [20] and is our preferred oligomer chemistry for in vivo evaluation in animal studies and ultimately, for clinical application. 

Unless directly coupled to a cell-penetrating agent, most AOs require a transfection reagent or protocol for efficient delivery into cultured cells, and in the case of splice switching compounds, be taken up into the nucleus where pre-mRNA processing occurs. Most commonly, a lipid-based solution is complexed with the AO prior to transfection [21], as the creation of a cationic lipid-complex capable of entering the cell via endocytosis vastly increases the AO uptake into the cell, compared to a gymnotic AO uptake. For screening of 2-OMe PS AOs, we typically use commercially available lipid transfection reagents, depending on the cell type. We have previously described transfection methods for efficient delivery of PMOs into cultured cells [22]. In addition to the transfection reagent or protocol, other factors that affect AO transfection efficiency in vitro include cell type and hence expression of the target gene, cell passage number and growth stage in culture, differentiation status, cell density, media and cell morphology during transfection, and AO chemistry and sequence composition. Here, we describe in vitro methods for AO design and the factors to consider during an initial screen of the most effective splice modifying antisense sequences using 2-OMe PS.

## 2. Results

### Guidelines for Developing Splice Switching AOs

 Step 1—Selection of Target Motifs

Once a target gene transcript and strategy is identified (e.g., exon skipping to remove a compromised exon from a disease causing gene, or disrupt expression of a target gene), open source web-based bioinformatics tools SpliceAid 2 [11] and Human Splicing Finder [12] may be used to identify predicted splice motifs, such as exon splice enhancers or exon splice silencers (Figure 1). Antisense oligomers complementary to the potential splice-associated motifs will then be synthesised as 2-OMe PS AOs. To induce exon skipping, the canonical acceptor and donor splice sites are obvious and well-defined targets, and since most exons are less than 200 bases in length, an additional three or four AOs, typically 20–25 nucleotides in length, provide reasonable coverage in the first pass.

We have reported a nomenclature system for the AOs according to gene, species, exon number, and annealing coordinates [23] (Figure 2). The nomenclature begins with the name of the transcript (e.g., survival of motor neuron 1; *SMN1*), then the species of the target mRNA (e.g., H: human or M: mouse), followed by the target exon number of the specified transcript and specification of an acceptor (A) or donor (D) site. The annealing coordinates are shown in brackets from the 5′ to 3′ position within the pre-mRNA transcript. The intronic bases are designated with a negative prefix (-) and the exonic position with a positive (+) symbol. The annealing coordinates are the positions of bases relative to the acceptor or donor sites of the reference transcript as denoted by National Center for Biotechnology Information and *Ensembl genome browser 96*. It is important to describe the reference transcript ID, especially in the case of targeting gene transcript isoforms that are composed of different numbers of exons. This AO nomenclature provides researchers with a unique designation, conveying the precise annealing coordinates of the targeted transcript that is particularly relevant when refining AO design by micro-walking around a responsive annealing site. Subtle shifts in AO annealing coordinates or length are immediately apparent and facilitates optimal AO design. 

 Step 2—Choice of Cell Type

When possible, screening and evaluation of splice switching AOs should be performed in cells expressing the target gene transcript and protein. GeneCards [24] offers a quick and convenient guide to cell and tissue-specific gene expression. However, certain disease-affected cell types—e.g., motor neurons or photoreceptors—may be difficult, impractical, or impossible to obtain and propagate from living patients. Consequently, it may be necessary to initiate AO design and evaluation in patient-derived lymphocytes or skin fibroblasts, and while not optimal, these cell types are relatively easy to obtain and culture. Although target gene expression may be relatively low in lymphocytes or fibroblasts, it may be sufficient for initial proof of concept studies at the RNA level. 

When studying most muscle diseases, patient-derived myoblasts would be the preferred material for study. However, obtaining muscle biopsies for the propagation of myogenic cells requires more invasive procedures compared to collecting a skin punch or blood sample. Unless elective surgery has been scheduled, working with healthy or patient-derived myogenic cells should only be considered after careful deliberation and consultation. Dermal fibroblasts can be propagated and subsequently induced into the myogenic lineage using MyoD expressing vectors, a common method routinely used to differentiate fibroblasts into myogenic cells [25,26,27]. 

In some instances, it is not necessary to use patient cells for the design of potentially therapeutic AOs. For example, the most common type of Duchenne muscular dystrophy-causing mutation is the genomic deletion of one or more exons, with subsequent disruption of the open reading frame. Since a normal exon flanking the frame-shifting deletion must be excised to restore the reading frame, the current suite of dystrophin exon skipping oligomers were designed against the normal dystrophin sequence and first evaluated in cells derived from healthy individuals [1,2,8]. Furthermore, AO optimisation in healthy cells sets a high standard, since the full complement of splicing factors are present in the context of the normal transcript, and the transcript induced by skipping a frame-shifting exon will be out of frame and hence subject to nonsense-mediated decay. While developing AOs to excise human dystrophin exon 8, compounds were first evaluated using healthy myoblasts, and clear differences in exon skipping efficiencies were readily evident. However, when these same compounds were tested in amenable patient-derived myogenic cells (e.g., missing exons 3-7 or 5-7), the distinction between poor and robust exon skipping AOs was much less evident [28]. For AOs designed to downregulate expression of a target protein by disrupting normal splicing, screening can again be undertaken initially in healthy cell lines. Once optimised, proof of concept studies can then be initiated in patient-derived cells for further validation and protein studies. 

For gene transcripts not expressed in either fibroblasts, lymphocytes or myoblasts, other commercially available cell lines, such as HEK293 (human embryonic kidney epithelium) or SH-SY5Y (neuroblastoma) lines available from repositories (Coriell Institute for Medical Research or American Type Culture Collection), may suffice. However, for specific disease-causing mutations, patient-derived cells are required for testing and validation of the AO and assessing the mutation specific effect. The construction of mini-gene assays to study the consequences of a particular splice mutation and AO intervention can be helpful. However, the utility of mini-gene assays can be limited by the length and structure of the cloned exonic and intronic sequences, and the cell type or strain used.

 Step 3—Delivery Reagents 

Once the AOs are designed and an appropriate cell type is chosen, we recommend exploring different transfection reagents for optimal AO delivery and uptake. Depending on the mechanism of action, AOs are required to be delivered to either the cytoplasm (for protein translation blockade and RNase H mediated mRNA degradation) or the nucleus (to alter pre-mRNA processing, including splicing or polyadenylation). Oligomers labelled with various fluorophores (e.g., FAM and TET) may be used to assess gross transfection efficiency, distribution, and uptake of each reagent; however, when exploring novel splice modification approaches, we recommend using a validated splice modulating AO as a control. As shown in Figure 3, one such control is an AO designed to induce skipping of exon 3 from the *ITGA4* transcript, a widely expressed gene in many different cell types. Not only will transfection of a control AO provide a guide to the transfection efficiencies, it can also be useful in assessing RNA quality and quantity in conjunction with the RT-PCR assays. 

The *ITGA4* transcript in healthy human fibroblasts was analysed by end point RT-PCR. The cells were transfected with 100 nM ITGA4 H3A (+ 30 + 49), a 2-OMe PS AO that induces skipping of exons 3 and 4 from the *ITGA4* gene transcript, using three different lipid-based transfection reagents (Figure 3A). Although all transfection reagents tested did deliver the AO, the transfection using Lipofectamine™ 3000 reagent induced the highest level of exon skipping in these cells. Lipofectamine™ RNAiMax showed a similar exon skipping pattern as Lipofectamine™ 3000, however, higher levels of cell death were induced by the former. The recommended lipid-based transfection reagents for delivery of 2-OMe PS AOs into the cell lines tested in our laboratory are listed in Table 1. Once the optimal transfection reagent is identified for a particular cell type, evaluation of AO sequences can proceed.

One important parameter to consider when designing primers for RT-PCR analysis of the full-length and AO-induced transcripts is to place the forward and reverse primers a few exons away from the targeted exon. We have now encountered several examples where targeting one exon for exclusion from the mature mRNA also influences recognition and retention of flanking exons and introns. As shown in Figure 3, amplification of the *ITGA4* transcript from exons 1 to 10 (Figure 3A) showed robust exon skipping, but not when amplified from exon 1 to 4 after transfection with the same AO designed to skip exon 3 (Figure 3B). 

 Step 4—Initial AO Screen

In early AO splice switching studies, the use of negative AO control sequences—either random, scrambled, or unrelated sequences—was essential to confirm specific target modification. Establishing target specificity is particularly crucial in situations where gene downregulation is the desired outcome. However, in many cases of splice switching, either exon skipping, exon retention or intron retention, the presence of a novel transcript is proof of the anticipated antisense mechanism. When AOs designed to a specific target do not affect the processing of that specific gene transcript, it is likely (but not inconceivable) that imperfect annealing to another pre-mRNA would have a minimal, if any effect.

Depending upon the gene and targeted exon, it has been our experience that up to two out of three AOs designed in a first pass can induce some level of exon skipping. However, targeting certain motifs noticeably results in more efficient exon skipping than others, and when developing any AO for clinical use, it is obvious that the most appropriate compound will be one that induces robust splice switching at a low concentration. The use of a positive transfection control AO is recommended for each transfection experiment, as this can control for transfection efficiencies across different experiments. It is also important to note that cell confluency, passage number, and other culture conditions can substantially influence transfection efficiency in primary cells and may lead to variations in AO efficacy between biological replicates. 

In some cases, individual AOs are ineffective at modifying exon selection, even after transfection at high concentrations. We have frequently found that selective AO cocktails, which include two or more AOs used in conjunction for a given exon target, mediate exon skipping in a synergistic manner, while each AO transfected alone is ineffective [18,29]. Conversely, we have also observed a marked decrease in exon skipping efficiency when two highly effective AOs are combined.

**Note**: It is recommended to confirm the identity of novel ‘exon skipped’ products by direct DNA sequencing, as nearby cryptic splice sites may be activated and generate amplicons of a similar length to the expected product. A difference of only a few bases in length can be difficult to resolve on an agarose gel, and such differences would be impossible to detect in longer RT-PCR products representing multiple exons [30].

Upon identification of amenable sites in the pre-mRNA that induce the desired splice modulation, AOs can be further optimised by ‘micro-walking’ and shifting the AO annealing sites in either direction to ensure the most amenable splice motifs have been targeted. An example of micro-walking is illustrated in Figure 4. Generally, to find the most effective AO, the annealing sites are moved five nucleotides in either the 5′ or 3′ direction, while retaining the same AO length. As shown in Figure 4, secondary screening of the AOs, ITGA4 H3A (+ 41 + 65) and ITGA4 H3A (+ 51 + 75), marginally improved exon skipping efficiency, compared to ITGA4 H3A (+ 46 + 70) indicating this general region would be suitable as an AO target. Alternatively, shifting the annealing site targeted by ITGA4 H3D (+ 6 − 19) further into the intron with ITGA4 H3D (+ 1 − 24) improved exon skipping efficiency from 18% to 34%. If considered necessary and of particular relevance, further micro-walking could be undertaken via moving the lead AO candidate target sequence by a few nucleotides in either the 5′ or 3′ direction. As a final optimisation step, once the most responsive or amenable annealing site is defined, the AO length may be truncated from either end. Shorter AOs are not only more efficiently synthesized, but substantially less costly to produce, an important consideration that will influence eventual clinical implementation. In some cases, AOs longer than 25 bases may be justified and must be considered on a case-by-case basis. We showed that efficient dystrophin exon 16 could be induced by overlapping 25 mers but increasing the length to a 30 mer resulted in a four-fold increase in exon skipping efficiency [31]. Hence, a 20% increase in AO length (and cost) resulted in a 400% increase in potency as assessed in vitro, thus allowing cost: benefits to be assessed. 

As a final evaluation to demonstrate reproducibility and efficacy, AO titrations should be performed to discriminate between AOs that induce similar levels of exon skipping, as shown in Figure 4B. Both ITGA4 H3A (+ 41 + 65) and ITGA4 H3A (+ 51 + 75) resulted in efficient exon skipping at 100 nM. However, when both AOs were transfected at 50 nM, the lower efficiency of ITGA4 H3A (+ 51 + 75) compared to ITGA4 H3A (+ 41 + 65) was evident.

## 3. Discussion

Here we describe general guidelines for developing, screening, and refining splice modulating AO sequences. It has been our experience that within a given gene transcript, some exons are readily excised from the mature mRNA, albeit at variable efficiencies in vitro, whereas other exons in the same transcript are more resistant to exon skipping [22,29]. In some cases, there may be either only a single region found to mediate AO-induced exon skipping, or combinations of AOs are needed to induce exon skipping, with the combinations sometimes acting synergistically [29]. We are yet to determine parameters that predict the ideal exon or transcript target for the design of splice modulating AOs, and experimental optimisation is therefore critical to the development of AO therapeutics. 

As with all forms of genetic therapies, AO delivery is a crucial aspect for splice switching efficacy. Relatively minor changes to the transfection protocol can dramatically improve or weaken apparent AO splice switching efficiency, and consequently, we suggest that before any AO is classified as ineffective in modulating splicing, various transfection methods should be evaluated. We describe an AO sequence that can be used as a positive control to modify the expression of the *ITGA4* transcript, widely expressed in most cell types. We suggest the use of this AO as a positive transfection control, an important inclusion during the initial stages of AO strategy design. A positive control AO allows one to monitor the experimental protocols (e.g., transfection, RNA extraction, RT-PCR). 

Oligomer synthesis demands high coupling efficiencies and hence oligomer length can be an important and significant consideration in AO drug design, highlighting the need to keep the AO length as short as possible while still maintaining adequate exon skipping levels. Some sequences can be challenging to synthesise, for example AOs with a high GC content (>75%) and sequences that include stretches of four Gs have the ability to form G-quartets [32]; structures that stack on top of each other and form tetrad-helical structures, thus severely inhibiting the functionality and solubility of the oligomer [33]. 

One crucial aspect of developing splice modulating AOs is to enhance stability in biological systems without compromising efficiency and introducing toxicity. At present, 2-OMe PS AOs are cost-effective for initial screening. However, in several studies, the oligonucleotides on a PS backbone have shown toxicity, off-target effects, and injection site reactions [34,35,36,37]. Nevertheless, these negatively charged AOs are ideal as research tools as they can be readily transfected into cultured cells as cationic lipoplexes. The lead AO sequences identified by 2-OMe PS AO screens may prove more effective when synthesised as the clinically safe PMO chemistry. There are no reports of serious adverse events occurring after long-term treatment with *Exondys 51* [1,2], while the same cannot be said for the 2-OMe PS AO drug *Drisapersen*. PMOs escape the electrostatic repulsion from the negatively charged RNA or DNA due to their neutral backbone [20,38]. This enables much higher binding specificity and affinity when compared to 2-OMe PS AOs. Furthermore, PMOs exert little to no off-target effects or non-specific binding, which is largely attributed to their neutral charge [20].

In conclusion, we have described a robust approach for developing and designing splice modulating AOs that may eventually enter the clinic. Extensive refining of the AO sequences to achieve the shortest oligomer with high efficacy is crucial to reduce the cost and production issues of the candidate oligo sequences. Antisense oligomer-mediated modulation of gene transcripts involved in genetic diseases has great potential for therapeutic application. The systematic evaluation of AOs in the manner we describe here will ensure the selection of the most efficacious and safe AOs for clinical trials.

## 4. Materials and Methods

### 4.1. Cell Culture

All cell culture reagents were purchased from Gibco, ThermoFisher Scientific, (Scoresby, Victoria, Australia) unless otherwise stated. Primary dermal fibroblasts obtained from a healthy volunteer, with informed consent (approved by the Murdoch University Human Research Ethics Committee, approval number 2013/156, 25 October 2013) were propagated in Dulbecco’s modified Eagle medium (DMEM) supplemented with L-Glutamine and 10% foetal bovine serum (FBS) (Scientifix, Cheltenham, Victoria, Australia).

### 4.2. Antisense Oligonucleotides (AOs)

Antisense oligonucleotides comprising 2′-*O*-methyl modified bases on a phosphorothioate backbone (2-OMe PS) were synthesised in-house on an Expedite 8909 Nucleic Acid synthesiser (Applied Biosystems, Melbourne, Victoria, Australia) using the 1 µmol thioate synthesis protocol as described previously [39]. Briefly, phenyl acetyl disulphide was used in the sulphrisation of the oligonucleotide. The 2’-hydroxyl positions are protected with t-butyldimethylsilyl group. Specifically, benzoyl was used as the protection reagent for nucleotide A and C monomers, while isobutyryl was used for nucleotide G monomer and U monomer not requiring protection. After synthesis, the oligonucleotides were cleaved from the support following incubation in ammonium hydroxide for a minimum of 16 h at room temperature. The AOs were subsequently desalted under sterile conditions using the NAP-10 columns (GE Healthcare, Sydney, NSW, Australia) according to manufacturer’s instructions. A list of 2-OMe PS AOs used in this study are summarised in Table 2. 

### 4.3. 2′-O-Methyl Phosphorothioate AO Transfection

Approximately 15,000 fibroblasts were seeded onto 24-well tissue culture plates and incubated overnight at 37 °C in fibroblast propagation media. When approximately 70–80% confluent, cells were transfected with 2-OMe PS AOs using Lipofectamine™ 3000, Lipofectamine™ RNAiMax, or Lipofectin™ transfection reagent according to the manufacturer protocols. Opti-MEM media was used for all transfections. Transfected cells were incubated at 37 °C for 24 h before RNA was extracted for transcript analysis.

### 4.4. RT-PCR

Total RNA was extracted using Trizol (Life Technologies, Scoresby, Victoria, Australia) according to manufacturer’s guidelines for transcript analysis. RT-PCR was performed using RNA (50 ng) from AO-treated and untreated cells and a Superscript III One-Step RT-PCR System (Life Technologies, Australia). Primer sequences for all RT-PCR primers used in this study can be found in Table 3. RT-PCR amplification was performed using the following thermocycling conditions: 55 °C for 30 min, 30 cycles of 94 °C for 30 s, 55 °C for 30 s, and 68 °C for 2 min. RT-PCR products were resolved on 2% agarose gels in Tris-acetate EDTA buffer and images were captured using a Fusion-FX gel documentation system (Vilber Lourmat, Eberhardzell, France). Product identity was confirmed by reamplification and purification of separated amplicons [40], followed by Sanger sequencing by the Australian Genome Research Facility (AGRF).

## Figures and Tables

**Figure 1 ijms-20-05030-f001:**
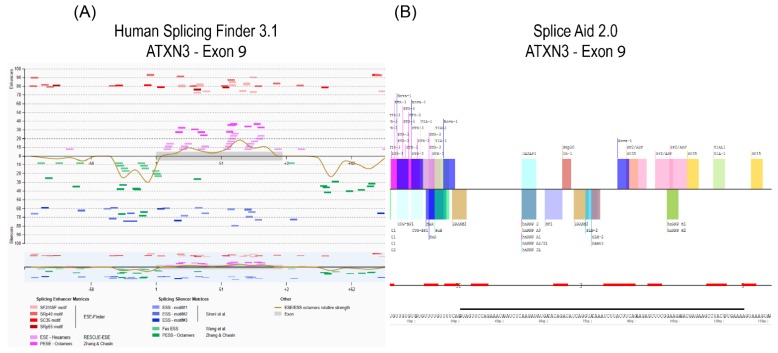
In silico prediction of splice motifs for exon 9 of the *ATXN3* transcript using Human Splicing Finder (**A**) and SpliceAid 2 (**B**). Exon position is identified by the black line above the sequence in SpliceAid 2.

**Figure 2 ijms-20-05030-f002:**
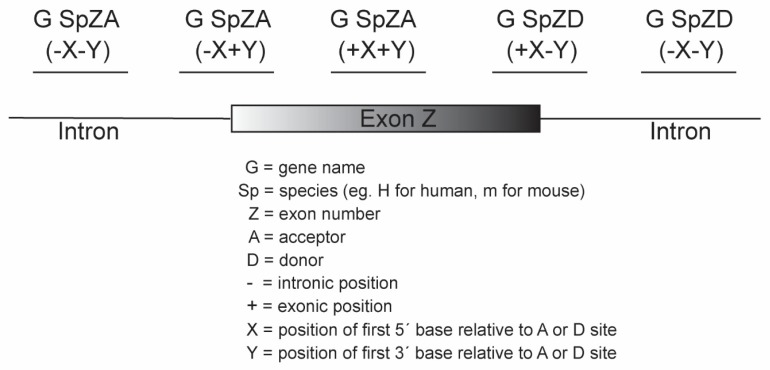
Nomenclature for antisense oligonucleotides.

**Figure 3 ijms-20-05030-f003:**
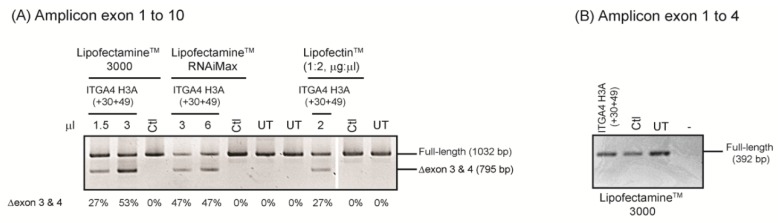
Modification of integrin alpha 4 (*ITGA4)* transcripts after healthy human fibroblasts were transfected with different lipid-based transfection reagents. (**A**) Three lipid-based reagents were used to transfect cells with ITGA4 H3A (+ 30 + 39) at 100 nM for 24 hr. Total RNA was extracted and RT-PCR was undertaken across exons 1 to 10 of the *ITGA4* transcripts. Transfection reagent volumes are indicated above the gel. Ctl; control AO that does not anneal to any sequence, UT; untreated, -; no template control. Percentages of *ITGA4* exon 3 and 4 skipping are indicated below the gel. The control AO was transfected using the maximum volume of transfection reagents. (**B**) RT-PCR amplification of the *ITGA4* transcript across exons 1 to 4 from RNA extracted from sample transfected with Lipofectamine 3000 from (**A**).

**Figure 4 ijms-20-05030-f004:**
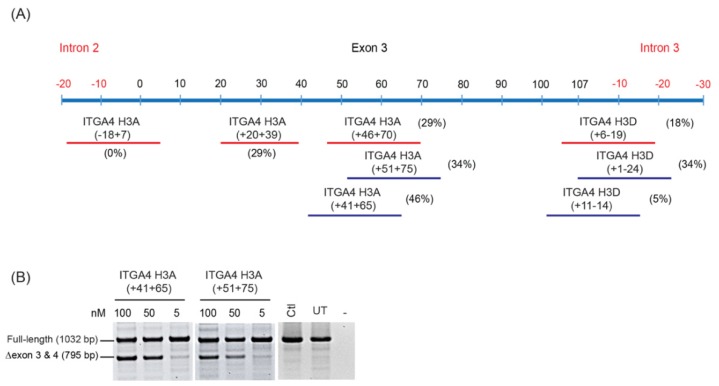
(**A**) RT-PCR analysis of *ITGA4* transcripts demonstrating refinement of splice switching AOs targeting *ITGA4* exon 3. Oligomers tested in the first screen are indicated by red lines and the micro-walked AOs tested in the second screen are represented by blue lines. Levels of exon skipping after transfection at 100 nM are indicated in brackets. (**B**) Comparison of two AOs that induce skipping of *ITGA4* exon 3 and 4 at various concentrations (100, 50, and 5 nM). Ctl; control AO that does not anneal to any sequence, UT; untreated, -; no template control.

**Table 1 ijms-20-05030-t001:** Recommended transfection reagents for different cell lines

Cell Lines	Transfection Reagents
Dermal fibroblasts	Lipofectin™, Lipofectamine™ 3000
Myoblasts and myotubes	Lipofectamine™ 2000
Lymphoblasts and lymphocytes	Nucleofection P3 Primary Cell Kit
Huh7	Lipofectamine™ 3000, Lipofectamine™ RNAiMax
HEK293	Lipofectamine™ 3000
H2k *mdx*	Lipofectin™
MO3.13	Lipofectamine™ 3000
iPSCs and neural stem cells	Lipofectamine™ Stem

Note: When evaluating AOs for splice modulation in primary cells, it is preferable to use cultures of lower passage number as we found that cultures with higher passage numbers tend to be transfected with lower efficiencies.

**Table 2 ijms-20-05030-t002:** List of AOs [41].

Name	Sequence (5′ – 3′)
ITGA4 H3A(+30+49)	UCUCUCUCUUCCAAACAAGU
ITGA4 H3A(-18+7)	GGGCUACCUAUAGCAUGUGAAAAUA
ITGA4 H3A(+20+39)	CCAAACAAGUCUUUCCACAA
ITGA4 H3A(+46+70)	GUGACCCCCAACCACUGAUUGUCUC
ITGA4 H3A(+41+65)	CCCCAACCACUGAUUGUCUCUCUCU
ITGA4 H3A(+51+75)	AAAGUGUGACCCCCAACCACUGAUU
ITGA4 H3D(+6-19)	GACCAGUUCCAAUACCUACCACGAU
ITGA4 H3D(+11-14)	GUUCCAAUACCUACCACGAUGGAUC
ITGA4 H3D(+1-24)	CUGUGGACCAGUUCCAAUACCUACC
Ctl	GGAUGUCCUGAGUCUAGACCCUCCG

**Table 3 ijms-20-05030-t003:** List of primers used in this study.

Name	Sequence (5′ – 3′)	Amplification Performed after Treatment with the Following AOs
ITGA4 ex1_F	gagagcgcgctgctttaccagg	All AOs
ITGA4 ex10_R	gccatcattgtcaatgtcgcca	
ITGA4 ex1_F	gagagcgcgctgctttaccagg	ITGA4 H3A(+30+49)
ITGA4 ex4_R	ggcactccatagcaaccacc	

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
