# Peer review of "Systematic Approach to Developing Splice Modulating Antisense Oligonucleotides"

_ijms, 2019, doi:10.3390/ijms20205030_

Round 1
Reviewer 1 Report
The manuscript reviews the methodology used in the development of antisense oligonucleotides directed to modulate splicing. It is a comprehensive review that describes the steps used in the laboratories of the authors to identify the best oligonucleotides for redirecting splicing for therapeutic uses. This field is a hot area of research and the methodology described could be of interest for a large number of biomedical researchers. The text is concise and well written.
There are only a very few minor suggestions to address.
The authors mention that oligonucleotide phosphorothioates were synthesized as described (ref 40). But neither ref 40 or the manuscript indicated the reagents used for the sulfurization nor the protection used for the O-methyl-RNA monomers. It will be nice to have these data in this manuscript as the oligonucleotides are used in cell culture after desalting.
At the end of the manuscript there are a few things that seems to be part of the template that have to be removed. Supplementary materials, there are no supplementary materials available. Abbreviations: LD, linear dichroism, TLA, DOAJ are not present in the manuscript.
Please check the abbreviation of the journal. Most of the names of the journals are full names but there are a few that are abbreviated. Some times even the full name is written in different form. For example Nucleic Acids Res (ref. 12) is written Nucleic acids research in refs 6, 13, 42 and Nucleic Acids Research in ref 9.
Author Response
Point 1: The authors mention that oligonucleotide phosphorothioates were synthesized as described (ref 40). But neither ref 40 or the manuscript indicated the reagents used for the sulfurization nor the protection used for the O-methyl-RNA monomers. It will be nice to have these data in this manuscript as the oligonucleotides are used in cell culture after desalting
Response 1: Referred to reference 41. Added requested information regarding synthesis. Lines 356 – 363
“Briefly, phenyl acetyl disulphide was used in the sulphrisation of the oligonucleotide. The 2'-hydroxyl positions are protected with t-butyldimethylsilyl group. Specifically, benzoyl was used as the protection reagent for nucleotides A and C monomers, while isobutyryl used for nucleotide G monomer and U monomer not requiring protection. After synthesis the oligonucleotides were cleaved from the support following incubation in ammonium hydroxide for a minimum of 16 hours at room temperature. The AOs were subsequently desalted under sterile conditions using the NAP-10 columns (GE Healthcare, St Giles, UK) according to manufacturer’s instructions.”
Point 2:At the end of the manuscript there are a few things that seems to be part of the template that have to be removed. Supplementary materials, there are no supplementary materials available. Abbreviations: LD, linear dichroism, TLA, DOAJ are not present in the manuscript.
Response 2:Removed sections of the template that are not pertinent to the current manuscript as requested by the reviewer.
Point 3: Please check the abbreviation of the journal. Most of the names of the journals are full names but there are a few that are abbreviated. Some times even the full name is written in different form. For example Nucleic Acids Res (ref. 12) is written Nucleic acids research in refs 6, 13, 42 and Nucleic Acids Research in ref 9.
Response 3:Have checked all abbreviations in references and made them uniform to reviewer’s request.
Reviewer 2 Report
The authors have submitted a very well written manuscript describing guidelines for developing splice modulating antisense oligonucleotides. The manuscript may potentially be accepted in the current form. My only minor comments are 1) Table 1 is the first to appear (in the introduction), but is placed between Figures 3 and 4; 2) in Supplementary Table 2, when denoting concentrations, commas should be corrected with dots (e.g., 0,3 mM).
Author Response
Point 1: My only minor comments are 1) Table 1 is the first to appear (in the introduction), but is placed between Figures 3 and 4.
Response 1: The authors have repositioned Table 1 to appear first. Table 1 is now located a Lines 209-212.
Point 2:Supplementary Table 2, when denoting concentrations, commas should be corrected with dots (e.g., 0,3 mM).
Response 2:We were unable to find supplementary Table 2 in the current manuscript. Would the reviewer be able to provide a location (line number) of the error so the authors can make the necessary amendment.